# Current Strategies for Modulating Tumor-Associated Macrophages with Biomaterials in Hepatocellular Carcinoma

**DOI:** 10.3390/molecules28052211

**Published:** 2023-02-27

**Authors:** Qiaoyun Liu, Wei Huang, Wenjin Liang, Qifa Ye

**Affiliations:** 1Zhongnan Hospital of Wuhan University, Institute of Hepatobiliary Diseases of Wuhan University, Transplant Center of Wuhan University, National Quality Control Center for Donated Organ Procurement, Hubei Key Laboratory of Medical Technology on Transplantation, Hubei Clinical Research Center for Natural Polymer Biological Liver, Hubei Engineering Center of Natural Polymer-Based Medical Materials, Wuhan 430071, China; 2The Third Xiangya Hospital of Central South University, Research Center of National Health Ministry on Transplantation Medicine Engineering and Technology, Changsha 410013, China

**Keywords:** TAMs, HCC, ROS, immunotherapy, biomaterials

## Abstract

Hepatocellular carcinoma (HCC) is the fourth most common cause of cancer-related deaths in the world. However, there are currently few clinical diagnosis and treatment options available, and there is an urgent need for novel effective approaches. More research is being undertaken on immune-associated cells in the microenvironment because they play a critical role in the initiation and development of HCC. Macrophages are specialized phagocytes and antigen-presenting cells (APCs) that not only directly phagocytose and eliminate tumor cells, but also present tumor-specific antigens to T cells and initiate anticancer adaptive immunity. However, the more abundant M2-phenotype tumor-associated macrophages (TAMs) at tumor sites promote tumor evasion of immune surveillance, accelerate tumor progression, and suppress tumor-specific T-cell immune responses. Despite the great success in modulating macrophages, there are still many challenges and obstacles. Biomaterials not only target macrophages, but also modulate macrophages to enhance tumor treatment. This review systematically summarizes the regulation of tumor-associated macrophages by biomaterials, which has implications for the immunotherapy of HCC.

## 1. Introduction

Primary liver cancer is a common malignancy worldwide, the majority of which is HCC, accounting for approximately 90% of primary liver cancers. In recent years, with a series of advances in tumor immunotherapy, immunotherapy for HCC has also received increasing attention [1]. HCC is an intractable malignancy with a variable and highly heterogeneous pathogenesis, making it difficult to achieve satisfactory results with a single treatment modality or drug [2]. Cancer immunotherapy is a potential treatment that uses a person’s immune system to fight cancer [3]. Compared to standard chemotherapy or radiation, it uses activated immune cells to target and destroy particular tumor cells. More importantly, immunotherapy has the potential to suppress and destroy metastatic cancers, which are a primary cause of cancer death globally [4]. Immunotherapy has become a new direction in the current research of HCC treatment [5,6]. Compared with traditional treatments such as surgery, chemotherapy, and radiotherapy, immunotherapy is relatively more specific and has fewer toxic side effects [1,7]. However, the liver has its unique immunosuppressive state, and no ideal antigenic target has been identified yet, so there has been no breakthrough in immunotherapy for HCC [5,7].

There are several immunosuppressive cells in the tumor microenvironment, including myeloid-derived suppressor cells (MDSCs), TAMs, and regulatory T lymphocytes (Tregs), which regulate immune escape, a key characteristic of malignant tumors [8]. In the TME, TAMs are the most numerous and plastic group of these cells [9,10]. They may be divided into M1-type TAMs, which are pro-inflammatory or anti-tumor, and M2-type TAMs, which are anti-inflammatory or pro-tumor, with M1-like macrophages being significantly less common than the M2 phenotype within the tumor [11,12]. To bridge innate and adaptive immunity and improve tumor immunotherapies, efforts have been made that reduce the number of M2-like macrophages or induce M2-to-M1 repolarization to offer significant potential [13]. Macrophages, one of the key elements of the innate system, can function as specialist phagocytes to protect against cancers. However, by expressing anti-phagocytosis proteins, tumor cells can conceal themselves to evade phagocytosis [14]. In controlling the tumor immunological microenvironment, macrophages are crucial. While M2 macrophages support tumor development, M1 macrophages prevent it. Treatment methods for solid tumors frequently involve the inhibition of M2-type TAMs and the repolarization of M2-type TAMs to M1-type TAMs. Moreover, the CD47-SIRP pathway has a major impact on these macrophages’ phagocytic activity against tumor cells that express CD47 since these macrophages express SIRP on their surface [15]. Therefore, inhibiting the CD47-SIRP pathway can further increase macrophages’ ability to fight tumors.

There has been significant interest in the recent use of nanotechnology in immunotherapy. Utilizing passive or active targeting effects, nanotechnology can be changed to accumulate only at tumor sites [12,16]. To prevent off-target negative effects and increase immunomodulatory efficiency, they offer enormous potential for directly controlling macrophage activities or indirectly delivering immunomodulators to reprogram TAMs [17,18,19]. Here, we will review the role of macrophages in anti-tumor immunotherapy, involving the depletion, repolarization of TAMs, and the blockage of the CD47-SIRP pathway to improve phagocytosis with cancer therapy, to clarify the significance of macrophages in tumor therapy.

## 2. Engineering Macrophages for Tumor Immunotherapy

### 2.1. Reprogramming TAMs

It has recently been shown that through the AMPK-NF-κB signaling pathway, metformin successfully changed the polarity of M2 phenotype TAMs to the M1 phenotype TAMs, preventing tumor growth and metastasis [20,21]. However, delivering metformin to M2 macrophages with pinpoint accuracy to significantly enhance cancer therapy results is still incredibly difficult. Yang’s group designed mannose-modified macrophage-derived microparticles (Man-MPs) loaded with metformin (Met@Man-MPs) to use the intrinsic cancer-targeting ability of macrophages and the high expression of mannose receptors in M2 TAMs (Figure 1A) [17]. This microparticle has the ability to specifically target M2-type TAMs and encourage their reversal polarization to M1-type TAMs with anti-tumor activity, thereby preventing tumor development. Met@Man-MPs can effectively improve the tumor immune microenvironment by promoting the recruitment of CD8^+^ T cells using cytokines such as TNF-α secreted by their polarization into M1-type TAMs and reducing the number of MDSCs and Tregs. Surprisingly, Met@Man-MPs contained matrix metalloproteinases that are produced from macrophages, which may efficiently break down the extracellular matrix of tumors and enhance the high infiltration of CD8^+^ T cells, as well as the enrichment and deep penetration of anti-PD-1 for tumor tissues. Furthermore, the same material (Met@Man-MPs) as above was prepared by Gan’s group and combined with Doxil for synergistic treatment [22]. Doxil induces immunogenic cell death (ICD) with cancer cells and stimulates the STING pathway, which makes it easier for microparticles repolarized M2-type TAMs to phagocytose tumor cells and activation of CD8^+^ T cells. Additionally, microparticles effectively boost drug tumor enrichment and penetration. Therefore, microparticles could treat cancer and produces potent anti-cancer ability as well as long-lasting anti-tumor immunological memory to prevent tumor recurrence. The results of this study demonstrate that microparticles are viable options to improve the chemoimmunotherapy effect of cancer therapy. Moreover, the other group using exosomes as carriers combined with pegylated oxide nanoparticles (loaded with chlorin e6) to improve immunity response in HCC by encouraging M1-type TAM polarization [23]. Exosome-specific markers CD9 and CD63 were expressed at significant levels. In vitro and in vivo, PION-contained exosomes can enhance M1 macrophage polarization in a dose-dependent manner. Notably, PION-contained exosomes were able to considerably increase the quantity of ROS produced by macrophages and greatly slow the development of tumors in mice receiving HCC xenografts.

Tumor hypoxia is one of the main causes of tumor treatment failure. There are few effective treatments for tumor hypoxia, and many findings suggest that the adaptive mechanism of tumor evasion for immune surveillance is also directly connected to the poor therapeutic impact of sorafenib [27,28,29]. Yu’s group developed an intelligent nanodrug PFH@LSLP that can target CXCR4, which is highly expressed under tumor hypoxic conditions, and deliver both sorafenib and CSF1/CSF1R inhibitor PLX3397 [30]. The liposome is based on an O_2_ supply of perfluoroohexane (PFH) core liposome and a surface linked CXCR4 antagonist LFC131 peptide. The PLX3397 activates the immune response by inhibiting the CSF1/CSF1R pathway in TAMs, further enhancing CD8^+^ T cell infiltration and reversing tumor immunosuppression (macrophage polarization). Anti-tumor effect on HCC mice and Patient-Derived Xenografts (PDX) of HCC patient origin suggests that nanoparticles can overcome drug resistance and synergistically alleviate tumor hypoxia, regulating drug resistance-associated genes, and remodeling the immune microenvironment.

Shan’s team developed a novel drug delivery system using Bi/Se nanoparticles (NPs), which were produced using a straightforward reduction reaction approach (Figure 1B) [24]. The findings revealed that synthesized nanoparticles did well in alleviating hypoxia and the immune suppression status of HCC after loading Lenvatinib (Len). The radiotherapy (RT) region may be precisely tagged following the injection of synthesized nanoparticles, according to in vivo CT imaging studies. Following the injection of synthesized nanoparticles, in vivo therapy was carried out using Len’s distinctive and durable features, resulting in good RT results in vivo. Synthesized nanoparticles may restructure and restore tumor blood vessels, reduce tumor hypoxia, increase the percentage of CD4^+^ and CD8^+^ T cells of tumors tissue, and induce macrophage polarization, according to extensive testing and histological stains. The remarkable use of synthesized nanoparticles as a potential therapeutic agent for HCC of RT is highlighted in this study. Meanwhile, the O_2_ supply microcapsules can dramatically improve radiation effectiveness in a mouse model of HCC [31]. Combining oxygen microcapsule therapy with radiation therapy lowers the quantity of TAMs while reshaping pro-tumor M2-type TAMs into anti-tumor M1-type polarization. This improves hypoxia in tumors, which in turn activates T cell-based anti-tumor immunity. According to the findings, RT coupled with oxygen microcapsule treatment, which decreases tumor hypoxia, is a potential approach to treating HCC. Guo’s group designed a metallodrug, lenvatinib was incorporated into mesoporous Fe_3_O_4_ nanoparticles and then bovine serum albumin (BSA) was conjugated to the surface of the nanoparticles (BSA-mFe@Len NPs) [32]. BSA permitted pH-responsive lenvatinib release of mesoporous Fe_3_O_4_ in acid TME. The injection of nanoparticles could improve the vascular normalization, recruitment CD4^+^ and CD8^+^ T cells, reduce the Tregs, induce cancer cell apoptosis, and alter the aberrant TME. To synergistically increase anti-tumor immunity, mesoporous Fe_3_O_4_ nanoparticles reprogrammed M2-type into M1-type after TAM phagocytosis. The metallodrug acquired an anti-tumor immune response as well as long-lasting immunological memory in the body, caused tumor vascular normalization effects, and significantly inhibited tumor development. Additionally, it performed well on magnetic resonance imaging, suggesting potentially therapy application.

The NanoMnSor nano-platforms, developed by Chen’s group, a tumor-targeted nanoparticle that effectively delivers drug and oxygen-producing into HCC [19]. This work discovered that nano-platforms provide stimulating responsive magnetic resonance imaging and sorafenib release characteristics following break down into Mn^2+^ ions in the tumor tissue, and then relieving tumor hypoxia. Additionally, MnO_2_-exposed macrophages showed elevated mRNA for the M1-type TAMs. This work demonstrates that therapy with nano-platforms dramatically reduces primary tumor development and distant metastasis, inhibits sorafenib-induced tumor vascularization, and prolongs the survival cycle of HCC mice. The synergistic treatment effectively increases CD8^+^ T cell levels at tumor sites, induces TAMs polarization, and reduces the hypoxia-induced tumor infiltration of TAMs, NanoMnSor also reprograms the immunosuppressive TME, improving the effect of anti-PD-1 and systemic anti-tumor immune response for the treatment of cancer. According to this study, oxygen-producing nanoparticles have the ability to deliver antitumor drugs, effectively alter the hypoxic TME, and overcome drug resistance caused by hypoxia.

In HCC, the large number of selectively activated macrophages (M2) in the tumor was associated with poor prognosis and failed immunotherapy with checkpoint blockade [33,34]. Zhang’s group found that synthetic high-density lipoproteins (sHDLs) preferentially deliver cargo to M2 and Hepa1-6 HCC cells, and then the authors designed a functional sHDL containing esterase-responsive prodrugs of vadimezan and gemcitabine (VG-sHDLs) (Figure 1C) [25]. Nanoparticles induced the release of damage-related molecular patterns and various cytokines from HCC cells, thereby promoting the differentiation of monocytes to M1-type TAMs and dendritic cells. Moreover, nanoparticles were able to kill M2-type TAMs without significant toxic effects on M1-type TAMs. The results shown that the intra-tumoral levels of M1-type TAMs and pro-inflammatory molecules are increased, while M2-type TAMs and the cytokine IL-10 are decreased after intravenous injection of VG-sHDLs. The above results increase the trigger and activation of CD8^+^ T cells, which eliminate mouse tumors and create an immunological memory in the body that inhibits tumor recurrence. This research fully reveals the interaction between high-density lipoproteins and cancer cells and gives a new theoretical basis for combining conventional chemotherapy and emerging immunomodulators for chemo-immunotherapy of HCC. 

Tumor microwave ablation therapy is only effective for tumors less than 3 cm in diameter, while it is ineffective for large-volume tumors and distal metastases [35,36]. To address this challenge, Liu’s team has recently developed a metal ion hydrogel based on sodium alginate (Figure 1D) [26]. It was found that this metal ion hydrogel can significantly improve the effect of microwave heat by the ion-limited effect and can effectively constrain the range of microwave heating. Meanwhile, after injection into the tumor site, the high concentration of Ca^2+^ in this hydrogel combined with mild heat therapy can induce immunogenic death of tumor cells found by disrupting intracellular Ca^2+^ homeostasis, which not only significantly improves its effect on local tumor ablation therapy, but also induces the anti-tumor immunity and induces the polarization of M2-type TAMs into M1-type TAMs, especially with Mn^2+^. The combination with Mn^2+^ (an activator of the STING pathway) is effective in suppressing distal tumors that are not directly treated and in inhibiting tumor recurrence. This work demonstrates that this metal–ionic hydrogel system can be used as both a microwave sensitizer and an immunostimulatory adjuvant to enhance microwave ablation therapy. Because the material system has good biosafety and components such as sodium alginate and calcium chloride are common clinical drug excipients, this treatment strategy has a high prospect of clinical translation. The man-IONPs nanoparticles were created by Liang’s team to reprogram M2-type TAMs into the anti-cancer M1-type TAMs [37]. Research performed showed that the macrophage infiltration was accelerated via inadequate microwave ablation, nanoparticles particularly targeting M2-type TAMs and aggregated in peri-ablation zones. The immunosuppressive milieu might be changed into an immune-activating one thanks to the nanoparticles’ simultaneous induction of reprogramming pro-tumor M2-type TAMs into anti-cancer M1-type TAMs.

To treat liver cancer, Zhang’s team created therapeutic nanoparticles (TNPs) that simultaneously delivered the drugs doxorubicin and IL-12 [38]. The synthesized nanoparticles displayed extended blood circulation, effective tumor accumulation, and tumor cell internalization. After that, in the tumor microenvironment, DOX and IL-12 simultaneously enter the tumor. In turn, macrophages could obtain new instructions thanks to TNPs’ locally responsive characteristics. More importantly, TNPs can effectively limit the development of the H22 tumor while having no evident negative effects. To further increase the anti-cancer effectiveness, synthesized nanoparticles can successfully reprogram macrophages toward the M1-type TAMs to lessen tumors. 

Zhu’s group used optogenetically modified engineered probiotic Escherichia coli (EcN) in combination with lanthanide upconversion nanomaterials (UCNPs) to develop a near-infrared light spatiotemporal response platform for cancer immunotherapy [18]. The UCNPs bound to EcN convert 808 nm near-infrared light to blue light, thereby activating the blue light response system in engineered EcN to induce the expression and secretion of flagellin B (flaB). flaB binds to Toll-like receptor 5 (TLR5) on the membrane surface of macrophages to induce the corresponding immune response for tumor therapy. In this study, the authors found that the intravenous injection of engineered EcN resulted in good intra-tumor colonization. After near-infrared light irradiation, the engineered EcN was able to produce and secrete flaB at tumor sites, successfully promoting TAMs polarization toward M1, activating immune responses, increasing the infiltration ratio of CD4^+^ as well as CD8^+^ T lymphocytes in the TME, and improving the immune-suppressive microenvironment of cold tumors, thus exerting anti-tumor effects. Compared with the control group, both liver and breast cancers were effectively inhibited in mice, and breast cancer metastasis was significantly improved in mice. This spatiotemporally controllable tumor bacterial therapy platform has good biosafety and is expected to provide new ideas for clinical tumor bacterial therapy.

### 2.2. Depleting TAMs

The malignant tumor HCC is resistant to both chemotherapy and immunotherapy [39]. Doxorubicin can cause immunogenic cell death (ICD) and is frequently used in arterial chemotherapy for HCC; however, its immunogenicity is still only moderate [40,41,42]. Huang’s group targeted poly(lactic acid)-hydroxyacetic acid copolymer (PLGA)–poly(ethylene glycol) (PEG)–aminoethyl ethanolamine nanoparticles to deliver icaritin and doxorubicin, which altered the immunosuppressed TME and sparked a strong immunological memory activation, effectively improving the early HCC effect in a mouse model [43]. It was found that icaritin induced mitochondrial autophagy and apoptosis in HCC cells, causing ICD. When the molar ratio of icaritin to doxorubicin was 1:2, there was a synergistic effect on the induction of ICD. In addition, synthesized nanoparticles combined with lenvatinib greatly extended the survival period in mouse with HCC [43]. The above results of this study reveal the HCC mechanism of icaritin by affecting mitochondrial autophagy, providing a successful immune-based treatment approach for HCC.

The cutting-edge treatment method for HCC is immune checkpoint blockade-based immunotherapy. Nevertheless, tumor immunological resistance and escape significantly limit the therapeutic efficiency of immune checkpoint blockade treatment [44]. Suo’s group designed manganese oxide-crosslinked bovine albumin/hyaluronic acid nanoparticles to loading of doxorubicin and indocyanine green (BHMDI) nano-composites (Figure 2A) [45]. The BHMDI nano-composites displayed dual-responsive pH/reduction drug release behavior and a high drug content (>46%). With catalytically converting intracellular H_2_O_2_ to O_2_, the nanoplatforms may effectively treat tumor hypoxia and vastly enhance the effectiveness of BHMDI-based photodynamic chemotherapy. The BHMDI nano-composites enhanced dendritic cell maturation, leading to the increase in CD4^+^ and CD8^+^ T cells, and simultaneously decreased the fraction of alternatively M2-type TAMs in malignancies. In addition to eliminating original tumors, anti-PD-1 conjunction with the nano-platforms prevented tumor recurrence, distant tumor development, and lung metastasis due to inducing potent global anti-cancer effects. The study demonstrated the viability of PD-1 blockade immunotherapy by potentiating it by correcting the hypoxic and immunosuppressive TME using photodynamic combined chemotherapy based on BHMDI.

Surgical resection is currently the most effective and widely used treatment modality to deal with HCC [46]. However, studies have found that the tumor recurrence rate 5 years after surgery is as high as 70%, and intraoperative bleeding and narrow margins are important risk factors for the recurrence of HCC after resection [47,48]. Therefore, intraoperative bleeding and narrow margins are clinical problems that need to be addressed to prevent the recurrence of HCC. NK cell permutation is considered promising tumor immunotherapy [49,50]. However, complex environments such as acid TME, neutrophil extracellular capture networks (NETs), and immunosuppression seriously affect efficacy [51,52,53]. To this end, Chen’s group developed a pH-responsive material containing a tumor acid neutralizer and NETs lytic enzyme (DNaseI) and combined it with NK cell percutaneous treatment to avoid tumor recurrence after narrow-margin surgery for liver cancer (Figure 2B) [54]. This material is able to be sprayed onto the surgical incision margin to generate an adhesive gel for fast hemostasis. In addition, it neutralized the acid microenvironment of the tumor, reduced local infiltration of immunosuppressive cells, and released DNaseI to degrade NETs. Animal studies demonstrated that this combination therapy significantly enhanced the efficacy of NK cell passaging without causing significant systemic toxicity. Furthermore, to treat HCC, Huang’s group created a nanoparticle called Nano-FdUMP (containing FdUMP (fluorouracil active metabolite)), a novel discovered compound created utilizing nanoprecipitation technology [55]. In HCC mouse models, combined effectiveness was accomplished. This was primarily due to the role that the nanoparticles played in the production of reactive oxygen species, which increased the effectiveness of ICD-based responses and cleared M2-type TAMs. 

**Figure 2 molecules-28-02211-f002:**
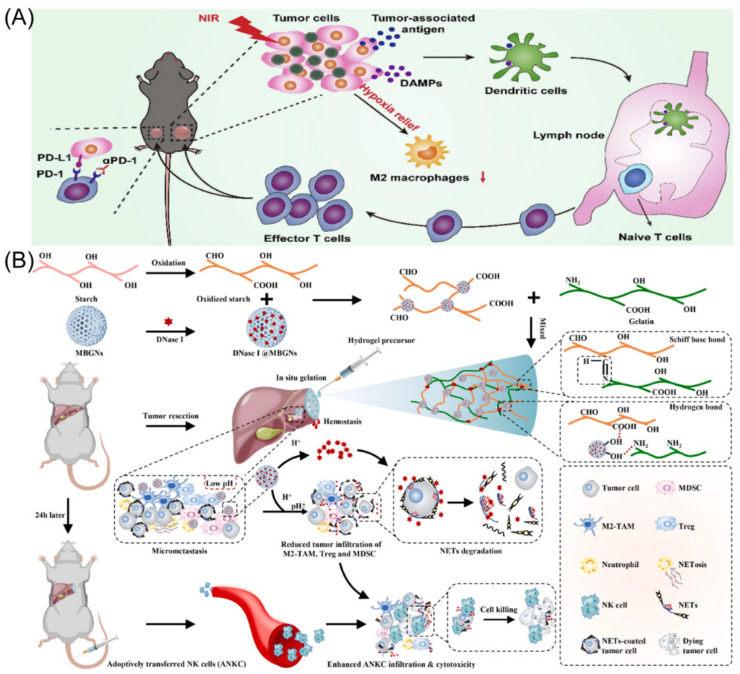
(**A**) The strategy through which BHMDI-induced ICD potentiates PD-1 blockade. Copyright 2022, Elsevier [45]. (**B**) Diagram of adhesive hemostatic gel preparation and combined treatment. Copyright 2022, Elsevier [54].

### 2.3. Ferroptosis and Polarization of Macrophages

xCT (encoded by SLC7A11) is crucial for cancer cell survival, proliferation, and malignant development and has been found to be highly elevated in many human tumors [56,57]. What should be noted is that the promotion of SLC7A11 overexpression on tumor growth and progression is undertaken in part by reducing ferroptosis [58]. Recent research has found that treatment techniques, which including immunotherapy and radiotherapy, trigger tumor cells ferroptosis to improve tumor suppression by altering SLC7A11 expression [59,60,61]. However, uncertainty exists about the precise effects of macrophage-mediated ferroptosis on TME, notably on TAM infiltration and polarization. Finding innovative and efficient anticancer treatments may be as simple as researching the regulating role of ferroptosis on TAMs for cancer therapy. 

The Man@pSiNPs-erastin prepared by Ji’s group particularly inhibits macrophage ferroptosis and polarization and enhances the anti-tumor effect when combined with anti-PD-L1 (Figure 3A) [62]. In this work, authors used mannose-functionalized porous silicon nanoparticles (Man@pSiNPs), a functional nanocarrier that can specifically target TAMs. Then, erastin was entrapped in Man@pSiNPs via mixing. It is important to note that a trustworthy indicator of tumor status is xCT expression, particularly in CD68^+^ macrophages, and a predictor of prognosis in HCC patients. Using an HCC model, the results determined that the nanoparticles knockdown in TAMs was effective in inhibiting tumor growth and metastasis via decreasing TAMs recruitment and infiltrating, suppressing TAMs polarization, and promoting ferroptosis function in TAMs (Figure 3B,C). In addition, xCT-mediated ferroptosis in macrophages significantly increased macrophage PD-L1 expression and improved the anti-cancer effectiveness of anti-PD-L1 treatment [62]. In conclusion, the results offer new perspectives on how to precisely avoid cancer by controlling ferroptosis activation in TAMs and controlling their recruitment and functional expression.

### 2.4. Enhanced Macrophage Phagocytosis

The CD47 self-marker binds to SIRPα and inhibits phagocytosis of normal cells, including red blood cells, by macrophages [63]. Additionally, CD47 shields tumor cells from macrophage phagocytosis, and high levels of CD47 expression correlate with a poor prognosis in several types of cancer [64,65]. Thus, it is expected that the blockade of the CD47-SIRPα axis restores the anti-cancer effect of macrophages [14].

Even though iron nanoparticles show the possibility of inducing M2-type TAMs to M1-type TAMs polarization, achieving the effective delivery of M2-type TAMs and Fe under cell uptake has been a major hurdle [66,67,68,69]. Gan’s group designed M2-type TAMs targeting peptide-linked Fe metal–organic frameworks (Dic@M2pep-Fe-MOF) to improve anti-tumor therapy (Figure 4A) [70]. Using nanoparticles, M2-type TAMs are effectively targeted and leakage through the hepcidin/ferroportin signaling pathway is reduced, thus leading to enhance M2-to-M1 macrophage repolarization (Figure 4C,D). More importantly, Dic@M2pep-Fe-MOF could effectively kill and phagocytose cancer cells, thus reconstituting the TME to form long-lasting protection for the body, resulting in a potent anti-tumor effect with the prevention of tumor recurrence (Figure 4B).

Glucose dysregulation and immune evasion are known to be two features of cancers that cause low therapeutic effectiveness and cancer recurrence [71]. To address the problems, Li’s group designed a unique nano-delivery system composed of the copolymer for the encapsulation of the clinical therapeutic drug CUDC101 and the photosensitizer IR780 (Figure 5A) [72]. The unique core–shell rod-like shape of the engineered copolymer gives TEMPO radicals to provide outstanding levels of magnetic resonance imaging. The labeling of the copolymer with glucose offers the copolymer excellent multimodal imaging properties. The effectiveness of CUDC101 and IR780 provide synergistic anti-tumor effects via photosensitizer-based photodynamic therapy and drug-induced CD47 suppression, demonstrating reprogramming to M1-type TAMs (Figure 5B,D). More interestingly, this research shows that the photodynamic therapy stimulation of p53 can also remodel TAMs, offering a potential combination of strategies for dual remodeling of the tumor microenvironment to exploit achieving a combined therapy to promote the change of the TME from a “cold” tumor to a “hot” tumor.

## 3. Conclusions

Engineering macrophages to regulate the immunosuppressive TME and the utility of M1-type TAMs in cancer therapy have been extensively discussed. The number of M2-type TAMs can be reduced by decreasing TAMs recruitment or depletion. Macrophages could be converted into pro-inflammatory M1-type TAMs by repolarization with various treatments. Together with the function of TAMs, their enhanced targeting capabilities synergistically improve tumor therapy efficacy. Despite considerable progress, there are still difficulties in using macrophages for medication delivery and cancer immunotherapy. First, due to the complexity of the TME and the continually shifting distribution of TAMs, it is unclear how macrophages are recruited and polarized in the TME. Second, macrophage-based anti-tumor immunity alone is insufficient to destroy cancer cause of the complex TME. When these issues are resolved, TAMs will be a powerful “weapon” that overcomes the difficulties now faced by immune modulation and solid tumor treatment, and it will show promise for clinical applications.

## Figures and Tables

**Figure 1 molecules-28-02211-f001:**
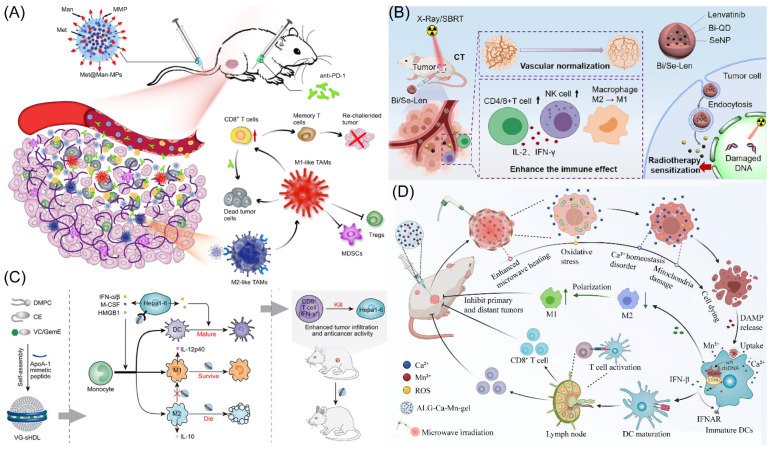
(**A**) Schematic representation of Met@Man-MPs acting as an effective drug to support anti-PD-1 treatment. Copyright 2021, Springer Nature [17]. (**B**) Schematic illustration of the radio sensitization mechanisms of Bi/Se-Len NPs in HCC. Copyright 2021, American Chemical Society [24]. (**C**) The design and function of sHDL are depicted schematically. Copyright 2021, Elsevier Ltd. [25]. (**D**) The diagram showing how the metallo-alginate hydrogel improves tumor microwave ablation therapy and enhances anti-tumor immunity. Copyright 2022, American Association for the Advancement of Science (AAAS) [26].

**Figure 3 molecules-28-02211-f003:**
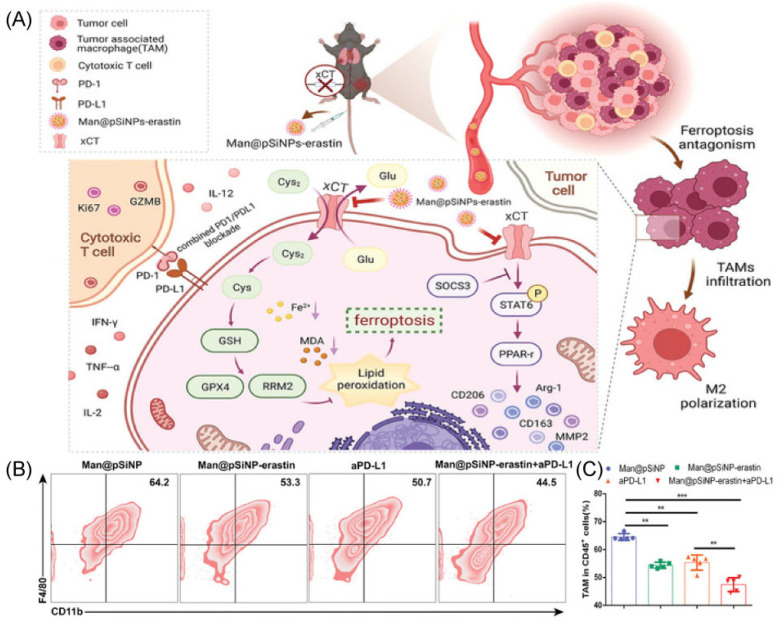
(**A**) The mechanism with Man@pSiNPs-erastin induced ferroptosis and polarization of macrophages. (**B**,**C**) Flow cytometry data shows the percent of TAMs in tumor tissues in response to different groups (*n* = 5). ** *p* < 0.01, *** *p* < 0.001. Copyright 2022, Wiley-VCH GmbH [62].

**Figure 4 molecules-28-02211-f004:**
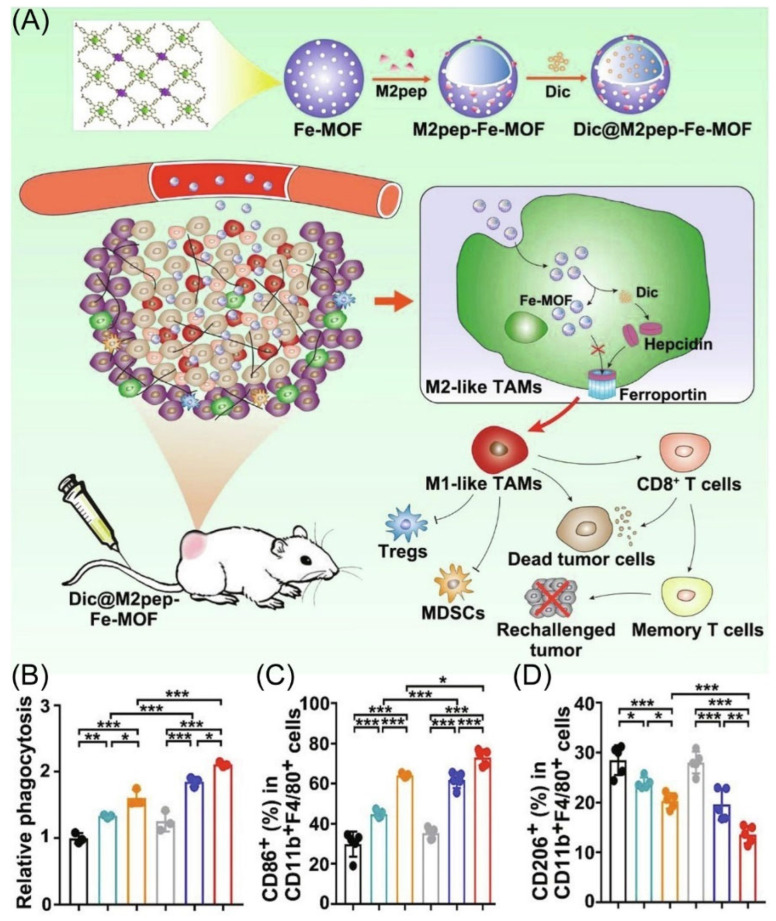
(**A**) Schematic diagram depicting materials to reshape the tumor immune microenvironment. (**B**) The phagocytosis ability of macrophage after different treatment co-culture under flow cytometry. (*n* = 3). (**C**,**D**) The percentages of M1 and M2 type TAMs in tumor tissue, respectively (*n* = 5). * *p* < 0.05, ** *p* < 0.01, *** *p* < 0.001. Copyright 2022, Elsevier [70].

**Figure 5 molecules-28-02211-f005:**
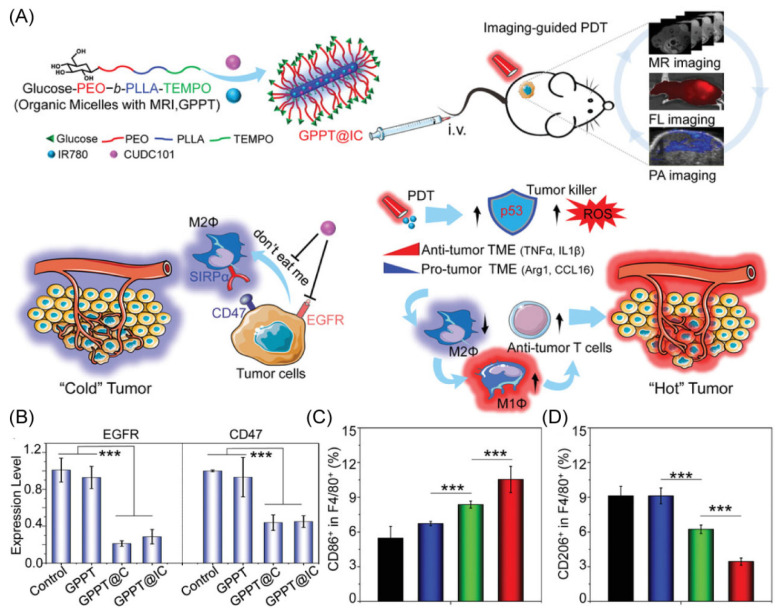
(**A**) Schematic diagram of the anti-tumor immune mechanism. (**B**) The RT-PCR was used to assess the suppression of EGFR and CD47 in Hep1-6. (**C**,**D**) M1-type TAMs and M2-type TAMs in the spleen of HCC mice under different treatments. *** *p* < 0.001. Copyright 2021, Wiley-VCH GmbH [72].

## Data Availability

Data are contained within the article.

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
