# Peer review of "Current Strategies for Modulating Tumor-Associated Macrophages with Biomaterials in Hepatocellular Carcinoma"

_molecules, 2023, doi:10.3390/molecules28052211_

Round 1
Reviewer 1 Report
The manuscript entitled “Biomaterials for Enhanced Immunotherapy of Hepatocellular Carcinoma via Modulate Tumor-Associated Macrophage” addresses important issues in the clinical management of hepatocellular carcinoma (HCC) and summarizes the novel therapeutic approaches based on the usage of nanoparticles. This narrative review is a qualitative synthesis of the results of previously published articles on the application of novel biomaterials targeting tumor-associated macrophages in the advanced therapeutic methods. Therefore, the topic of this manuscript is interesting and within the scope of the journal.
The manuscript is poorly written with major grammar and syntax errors (even in the Title), which make it very hard to read and understand. It is also poorly structured and there are some major issues that need to be addressed:
- For all the figures, it is stated that the images were copied from MDPI, but all the referenced articles for the images were published by other publishers (Wiley, Elsevier, etc.). This raises the question about the necessary permissions for reproduction, which should be obtained by the authors, or the figures need to be redrawn. The author rights and copyright need to be resolved.
- The image captions should include more details. For instance, the caption for Figures 2A and 3A states: “The basic method and mechanism of this research”, without specifying the referenced article and without any further description. Figures and the corresponding captions should “stand alone”.
- The first sentence in the Abstract section (line 13) contains “still” which should be deleted.
- Sentence in the lines 22 and 23 needs to be rephrased, since it is meaningless.
- Statement in lines 55-58 just repeats the previously stated claims in lines 47-51.
- Most of the microparticles are stated with their abbreviated names, without clarifying their exact chemical nature and constituents. Also, Met@Man-MPs are described in line 89, while the same microparticle was previously mentioned in the line 79. The authors write about the microparticles assuming that the common reader of the manuscript is familiar with all the mentioned nanomaterials and with the cited references. Without specifying that Gan’s team used Met@Man-MPs in combination with doxil, the statement in lines 90-96 about the enhanced effect of doxil are confusing and poorly connected with previous text.
- Sentences about the diameters of PIONs@E6, solubility and exosome markers are irrelevant. The usage of this method and the design of the cited study are poorly described and the section that refers to ref 21 is confusing.
- Sentences in lines 164-166 and 170-172 need to be rephrased in order to be meaningful.
- Line 179 – should “effective” be “ineffective”?
- Delete “can” in line 185.
- HBO-enhanced PD-1 antibody therapy is immunotherapy which targets TAMs, but is irrelevant for the topic of this article, since it does not include the usage of advanced biomaterials.
- “Synthesis nanonaparticles” is a syntagm used many times in the manuscript. “Synthesis” should be corrected with “synthesized”, since it is used as an adverb.
- The section from lines 231 to 250 should be separated in a new paragraph. There is also a repeated section in the sentence in lines 238 to 240. In line 240, “we found” needs to be replaced with “the authors found”, or with another appropriate term.
- “One of the WORST cancers” needs to be replaced with another appropriate term (line 258, page 6).
- Sentence in lines 272-273 needs to be rephrased and corrected.
- BHMDI should be defined. Also Nano-FdUMP require further explanation about the characteristics and the therapeutic mechanisms.
- Abbreviation GODM in the caption of Figure 2 should be defined.
- The first sentence in 2.33 Section needs to be corrected in order to be meaningful.
- It should be stated that Man@pSiNPs-erastin targets xCT. Otherwise, there is a missing link between the effects of this nanoparticle and the ferroptosis and the molecular mechanism is poorly described.
- Section 2.4. should be merged with section 2.1., since all these biomaterial described in Section 2.4. act on TAM reprogramming. The significance of CD47 is described for the first time at the beginning of section 2.4., but this receptor was previously mentioned in line 58. Also, this first paragraph is irrelevant for later paragraphs.
- The sentence in lines 342-343 is merely a repetition of previous statements.
Author Response
Dear Editor,
Thank you very much for your letter and comments from reviewers. I am very grateful for your comments for the manuscript entitled “Biomaterials for Enhanced Immunotherapy of Hepatocellular Carcinoma via Modulate Tumor-Associated Macrophage” (Manuscript ID: molecules-2221072). Based on the reviewers’ comments and requests, we have made relevant modification on the original manuscript, which are highlighted in red in the revised manuscript text.
The specific points of the reviewers are addressed on the following pages. We believe that the revised manuscript has now addressed the concerns raised by the reviewers and should be suitable for publication.
We would like to express our great appreciation to you and reviewers for comments on our paper. Looking forward to hearing from you on the final status of our manuscript soon.
Very Truly Yours,
Qifa Ye

Reviewer 2 Report
Reviewer’s Comments:
The manuscript “Biomaterials for Enhanced Immunotherapy of Hepatocellular Carcinoma via Modulate Tumor-Associated Macrophage” is a very interesting work. In this work, Hepatocellular carcinoma (HCC) is still the fourth most common cause of cancer-related deaths in the world. However, there are currently few clinical diagnosis and treatment options available, and there is an urgent need for novel effective approaches. More research is being paid to immune-associated cells in the microenvironment because they play a critical role in the initiation and development of HCC. Macrophages are specialized phagocytes and antigen-presenting cells (APCs) that not only directly phagocytose and eliminate tumor cells, but also present tumor-specific antigens to T cells and initiate anticancer adaptive immunity. The results are consistent with the data and figures presented in the manuscript. While I believe this topic is of great interest to our readers, I think it needs major revision before it is ready for publication. So, I recommend this manuscript for publication with major revisions.
1. In this manuscript, the authors did not explain the importance of the Immunotherapy in the introduction part. The authors should explain the importance of Immunotherapy.
2) Title: The title of the manuscript is not impressive. It should be modified or rewritten it.
3) Correct the following statement “This review systematically summarizes recent advances in the use of biomaterials to target and modulate the anti-tumor function of macrophages, which are important and recommended for the development of macrophage-based immunotherapy of HCC”.
4) Keywords: There so many keywords and reduce them up to 5. So, modify the keywords.
5) Introduction part is not impressive. The references cited are very old. So, Improve it with some latest literature like 10.1016/j.chemosphere.2022.133772, 10.1016/j.eurpolymj.2021.110783
6) The authors should explain the following statement with recent references, “In addition, synthesis nanoparticles combined with lenvatinib greatly extended the survival period in mouse with HCC”.
7) Add space between magnitude and unit. For example, in synthesis “21.96g” should be 21.96 g. Make the corrections throughout the manuscript regarding values and units.
8) The author should provide reason about this statement “In addition, xCT-mediated ferroptosis in macrophages significantly increased macrophage PD-L1 expression and improved the anti-cancer effectiveness of anti-PD-L1 treatment”.
9) Comparison of the present results with other similar findings in the literature should be discussed in more detail. This is necessary in order to place this work together with other work in the field and to give more credibility to the present results.
10) Conclusion part is very long. Make it brief and improve by adding the results of your studies.
11) There are many grammatic mistakes. Improve the English grammar of the manuscript.
Author Response

(The authors gave the same response as above.)

Round 2
Reviewer 1 Report
The authors made corrections according to my suggestions and significantly improved their manuscript.